# Targeting the HIV-infected brain to improve ischemic stroke outcome

Luc Bertrand[1], Fannie Méroth[1], Marie Tournebize[1], Ana Rachel Leda[1], Enze Sun[1] & Michal Toborek[1]

HIV-associated cerebrovascular events remain highly prevalent even in the current era of antiretroviral therapy (ART). We hypothesize that low-level HIV replication and associated inflammation endure despite antiretroviral treatment and affect ischemic stroke severity and outcomes. Using the EcoHIV infection model and the middle cerebral artery occlusion as the ischemic stroke model in mice, we present in vivo analysis of the relationship between HIV and stroke outcome. EcoHIV infection increases infarct size and negatively impacts tissue and functional recovery. Ischemic stroke also results in an increase in EcoHIV presence in the affected regions, suggesting post-stroke reactivation that magnifies pro-inflammatory status. Importantly, ART with a high CNS penetration effectiveness (CPE) is more beneficial than low CPE treatment in limiting tissue injury and accelerating post-stroke recovery. These results provide potential insight for treatment of HIV-infected patients that are at risk of developing cerebrovascular disease, such as ischemic stroke.

---

[1] University of Miami Miller School of Medicine, Department of Biochemistry and Molecular Biology, Miami, FL 33136, USA. Correspondence and requests for materials should be addressed to L.B. (email: l.bertrand@med.miami.edu) or to M.T. (email: mtoborek@med.miami.edu)

Before the era of anti-retroviral drugs (ARVd), HIV infection was a deadly disease that was characterized by a slow progression to AIDS and death, often from co-infection due to an immunocompromised status. The development of ARVds and their usage in antiretroviral therapy (ART) changed the disease outlook to a chronic condition. While the replication of the virus is controlled by ART, it is not eliminated and the patients have to continue therapy for the rest of their life. HIV can enter a state of latency in multiple cells throughout the body. Viral reservoirs are mainly present in T cells and macrophages; however, several cells of the central nervous system (CNS) have also been identified as potential reservoirs, such as perivascular macrophages, astrocytes, and even pericytes[1–7]. While most currently used ARVds prevent new cells from being infected, the cells that already harbor HIV genomes can continue to produce toxic viral proteins[8].

The survivability of HIV infection results in the aging of a seropositive population. Indeed, the life expectancy of a 20-year-old HIV-positive adult on ART is expected to be ~70 years[9,10] and ~70% of adults with HIV in the US are likely to be 50 or older by the year 2020[11]. While the virus is suppressed, and immune status is maintained, these patients continue to display high levels of co-morbidities such as neurodegeneration, metabolic, cardiovascular and cerebrovascular diseases[12–16]. Even though the exact mechanisms of HIV-related co-morbidities remain unclear, an increasing body of evidence demonstrates that both the toxicity of ARVds and low-level HIV activity are likely to be implicated[17–24]. Indeed, a low level of viral activity that persists in HIV-suppressed patients can lead to increased basal inflammation, apoptosis, and disruption of neuronal signaling[25,26].

Several epidemiological studies evaluated the mortality cause of HIV patients, and cerebrovascular disease (CVD) is among the most prevalent[27–30]. Interestingly, HIV infected patients with CVD are often younger and less likely to have predisposing conditions such as high-blood pressure and elevated cholesterol[12,27,31–33]. Multiple factors, linked to the HIV infection, could contribute to this increased susceptibility to CVD, including opportunistic infections, endocarditis, cachexia, coagulation abnormalities, dyslipidemia, and ART[31,32,34–38]. It has also been shown that HIV and its proteins can interact directly with the endothelium and contribute to increased incidence of atherosclerosis, a major contributing factor to cardiovascular disease[39,40]. Data obtained from post-mortem human brain samples demonstrate that even in virally suppressed patients, evidence of vasculopathy is present[41,42]. HIV infection can cause small vessel disease resulting in improper regulation of blood flow[43,44]. In addition, HIV can induce atherosclerosis, causing thickening of the artery wall, and thus restricting blood flow[45,46]. Finally, vascular inflammation associated with HIV infection can further compromise normal vascular functions[47–49].

In ischemic stroke, the interruption of cerebral blood flow deprives the brain tissue of essential nutrients and oxygen. This process results in a decrease in neuronal viability and triggers a pro-inflammatory response, which affects the microvasculature[50,51], causing loss of the integrity of the neurovascular unit (NVU), characterized by an increase in blood–brain barrier (BBB) permeability and disruption of both the basal lamina and tight junctions[52,53]. An increase in expression of cellular adhesion molecules and regulatory cytokines is also observed, leading to tissue infiltration with leukocytes which can mediate a secondary round of microvessel obstruction, edema formation, cellular necrosis, and tissue injury[51,54–57],. These events can occur as quickly as 30 min after the original occlusion. Days after stroke, the immune response gradually switches to an anti-inflammatory processes, guided by regulatory T cells, but also aided by the localized production of TGF-β and IL10 by microglia, astrocytes and macrophages[58,59]. Several steps of this process have been shown to be viable targets to reduce stroke burden, such as inhibiting adhesion molecule expression, immune cell activation, or increasing anti-inflammatory response[60,61].

The barrier function of the BBB has proved to be a challenge to the efficient treatment of several CNS diseases. From brain tumors to brain infections, the inability of therapeutic molecules to cross this restrictive interface creates a critical obstacle to treatment efficacy[62–64]. While ARVds can be present at effective concentrations in the periphery, their limited entry into the brain can result in their suboptimal concentrations in the CNS, leading to only partial inhibition of viral replication and favoring the formation of drug resistant mutations[64,65]. Restricted entry of ARVds into the brain depends on several characteristics of the drugs, such as size, charge, and interactions with BBB efflux pumps[64].

Several ARVds have been evaluated for their ability to reach the CNS, ranking their CNS penetration efficiency (CPE) on a scale from low (1 CPE) to high (4 CPE)[66]. The cumulative score of drugs used in ART gives an indication of efficacy in treating HIV in the CNS from low (<8 CPE) to high (>8 CPE). There is an ongoing discussion around what ARVds are optimal for treatment of HIV infection in the brain. From one side, a better penetration of ARVds into the brain may be beneficial for controlling brain infection; however, these drugs may also exert more prominent side effects. On the other hand, treatment with low CPE regiment has been associated with a higher incidence of HIV-associated neurocognitive disease and CSF escape, indicating that targeting HIV in the CNS may be more important than the side effects of ARVds[67–69].

While the relationship between HIV and stroke has been demonstrated in several publications using epidemiological data, no in depth studies of how HIV infection affects cerebrovascular disease have been conducted. In the current manuscript, we are reporting in vivo evidence of a direct impact of HIV infection on ischemic stroke outcome. The combination of a mouse model of HIV infection and the middle cerebral artery occlusion (MCAO) model enabled us to directly evaluate post-ischemic stroke injury progression and recovery in HIV-infected brain. Our results demonstrate that HIV infection results in a significant increase in ischemic stroke severity by affecting the integrity of the BBB and enhancing the inflammatory response. Importantly, we provide evidence that ART with a high CPE score provides a benefit to post-stroke tissue and function recovery of HIV-infected brain as compared to ART with a lower CPE score.

## Results

**EcoHIV increases infarct size and delays tissue recovery.** To evaluate the impact of EcoHIV infection on infarct size, C57BL/6 mice were infected with 1 μg of HIV p24 delivered through the left carotid artery using the ICA injection delivery method[70]. This method increases delivery of the virus into the brain and establishes low level infection in the CNS, typical of current HIV epidemic in humans. Mock infected mice were subjected to the same procedure; however, they were infused with saline. Three weeks post infection, ischemic stroke was induced by the middle cerebral artery occlusion (MCAO) method using a silicone coated suture to occlude the MCA for 60 minutes[71]. At 1, 4, 7 and 14 days post-stroke, brains were harvested, sliced using a 1 mm brain matrix and stained with 2,3,5-triphenyltetrazolium chloride (TTC). Representative images of brain lesions 24 h post ischemic stroke in mock and EcoHIV-infected mice are shown in Fig. 1a. A significant increase in infarct size was observed in EcoHIV-infected mice as compared to mock-infected animals at days 1 and 7 (Fig. 1b). A reduction in infarct size occurred at days 7 and

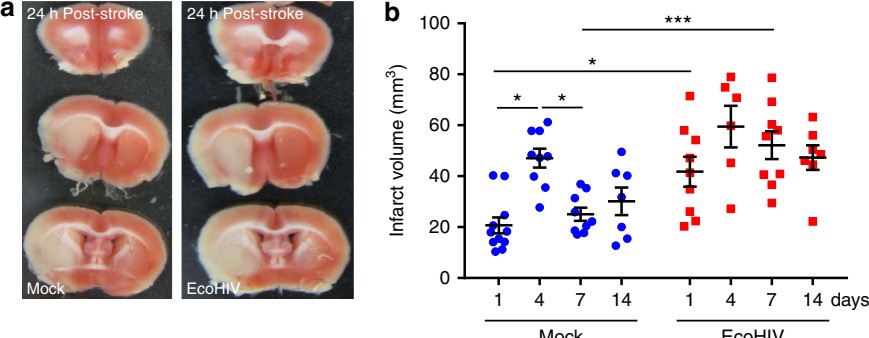

**Fig. 1** EcoHIV increases infarct size and delays tissue recovery. Mice were infected with EcoHIV (1 μg of HIV p24), and ischemic stroke was induced 3 weeks post infection by the occlusion of the middle cerebral artery. **a** TTC staining of brain sections of mock- and EcoHIV- infected mice 24 h post stroke. White areas denote tissue damage. **b** Calculated infarct volume for days 1, 4, 7 and 14 post-stroke in either mock- or EcoHIV infected mice; $n = 6–12$ mice per group, 3 independent experiments. Data presented as mean and SEM with individual data points. Source data are provided as a Source Data file. $*p < 0.05$; $**p < 0.01$; one-way ANOVA, followed by Tukey multiple comparison test

14 in mock-infected mice when compared to day 4, indicating tissue recovery. Representative images of brain lesions at these time points are illustrated in Supplementary Fig 1. However, such recovery was not observed in EcoHIV-infected mice. These results indicate that EcoHIV infection exacerbates infarct size and delays spontaneous tissue recovery.

**EcoHIV disrupts the NVU and enhances inflammatory responses**. To dissect how EcoHIV may impact early stroke disease process, we first evaluated the status of the BBB after infection. Animals were either mock or EcoHIV-infected and the BBB was evaluated 3 weeks post-infection. Because disruption of tight junction proteins and the extracellular matrix (particular laminin) can impact the BBB integrity and the outcomes of CVD[72], ZO-1 and laminin staining in brain microvessels were assessed on brain sections. Expression of ZO-1, a tight junction protein important for the maintenance of the BBB, was significantly reduced in EcoHIV-infected mice (Fig. 2a). In contrast, mean fluorescence intensity (MFI) of laminin immunoreactivity co-localized with CD31 (marker of endothelial cells) was not altered (Fig. 2b). Another factor that can influence CVD and BBB integrity is inflammation within the brain tissue and NVU[73]. An important indicator of such status is increased expression of cell adhesion molecules in the brain vasculature. The role of these molecules in the inflammatory process and ischemic stroke is related to the recruitment of pro-inflammatory cells into the brain. However, high levels of such molecules can decrease blood flow to the injury site through a phenomenon called cerebral no-reflow[74]. Therefore, we evaluated the levels of ICAM-1 (Fig. 2c) and P-selectin (Fig. 2d). Both adhesion molecules were elevated in EcoHIV-infected brains as compared to mock-infected controls. Microvessels delineation for quantification was performed using CD31 as a brain endothelium marker (Supplementary Fig. 2).

To evaluate BBB permeability, mice were injected with sodium fluorescein (NaF), and penetration of this marker into the brain parenchyma was measured as previously described[75]. There was a significant increase in NaF in the brain tissue in EcoHIV-infected mice as compared to the mock group, indicating disruption of BBB integrity (Fig. 2e). Overall, the results in Fig. 2 indicate that following infection with EcoHIV, the BBB is disrupted, placing the NVU in a proinflammatory state that may predispose the brain tissue to more severe stroke injury.

**EcoHIV diminishes post-ischemic stroke NVU recovery**. Several mechanistic events related to BBB integrity and function were evaluated post ischemic stroke in both non-infected and HIV infected brains. Because laminin levels were previously demonstrated to be increased in damaged tissue in order to restore BBB integrity[76], the initial series of analyses was focused on this protein. We observed a prominent elevation of laminin expression following ischemic stroke, however this increase was significantly diminished in EcoHIV infected mice as compared to mock-infected animals (Fig. 3a). In addition, immunoreactivity of the adhesion molecules ICAM-1 and P-selectin were significantly more increased after ischemic stroke in infected brains as compared to mock (Fig. 3b and c). ICAM1 protein expression levels were also quantified by immunoblotting at 1, 4, 7, and 14 days post ischemic stroke (Fig. 3d). Consistent with immunoreactivity results, EcoHIV markedly increased ICAM1 expression as compared to mock-infected mice.

Because adhesion molecules modulate recruitment of inflammatory cells, we next evaluated the infiltration of neutrophils by evaluation of Ly6g positive cells in the infarct area. The Ly6g immunoreactivity was more abundant in EcoHIV-infected brains as compared to mock-infected brains (Fig. 3e, left panel). Quantification of these results revealed that infiltration neutrophils 24 h post-ischemic stroke more than doubled in EcoHIV-infected brains compared to mock controls (Fig. 3e, right panel). Overall, the results in Fig. 3 demonstrate that while ischemic stroke alters the architecture and inflammation of the BBB, infection with EcoHIV can further compromise cerebral vascular responses and exacerbate tissue damage.

**Prolonged post-stroke inflammation in EcoHIV-infected brains**. Figure 1 demonstrates the differences in the initial infarct size and in the recovery process between non-infected and EcoHIV-infected mice, with the infected mice exhibiting larger tissue injury. In order to determine whether the changes in inflammatory profile correspond to tissue recovery, we compared mRNA levels of major inflammatory mediators by qPCR between the mock and EcoHIV infected mouse brain tissue 7 days post-stroke. mRNA expression of IL1β and TNFα did not show any significant differences between the mock + stroke and the EcoHIV + stroke groups (Fig. 4a, b). However, changes were observed in the expression of chemokine CXCL1 (Fig. 4c), which is involved in the recruitment of neutrophils, monocytes and macrophages. While the differences in CCL2 expression were not significant (Fig. 4d), they exhibited a strong tendency to be upregulated in the EcoHIV-infected as compared to mock-infected brains. Furthermore, the markers of astrocyte activation, GFAP (Fig. 4e), and monocyte/macrophage activation, Iba1 (Fig. 4f), were elevated in infected brains.

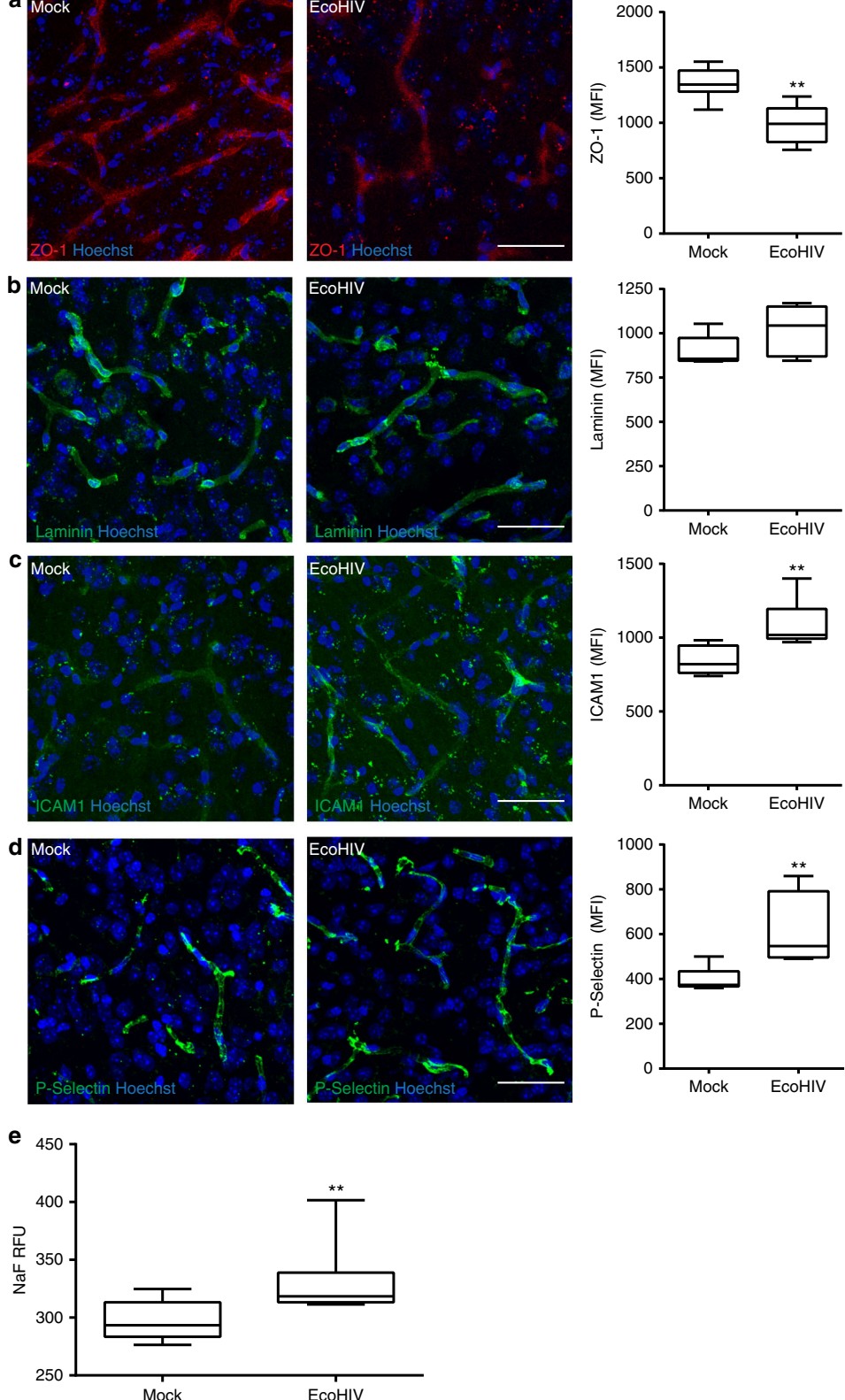

**Fig. 2** Disruption of NVU and inflammatory responses in EcoHIV-infected brains. Mice were infected as in Fig. 1, and analyses were performed 3 weeks post infection. Brain sections were stained for ZO-1 (**a**), laminin (**b**), ICAM-1 (**c**), and P-selectin (**d**). In **a**, **b**, $n = 6$ mice, 20 microvessels per mice, 2 independent experiments. In **c**, **d**, $n = 6$ mice per group, 10 microvessels per mice, 2 independent experiments. **e** BBB permeability in mock- and EcoHIV-infected mice as assessed by the sodium fluorescein (NaF) extravasation method; $n = 10$ per group, 2 independent experiments. Whiskers-box plots represent centerline median, with interquartile range and min-max whiskers. Source data are provided as a Source Data file. **p < 0.01 vs mock; unpaired $t$ test. **a**–**d** Scale bars: 40 μm

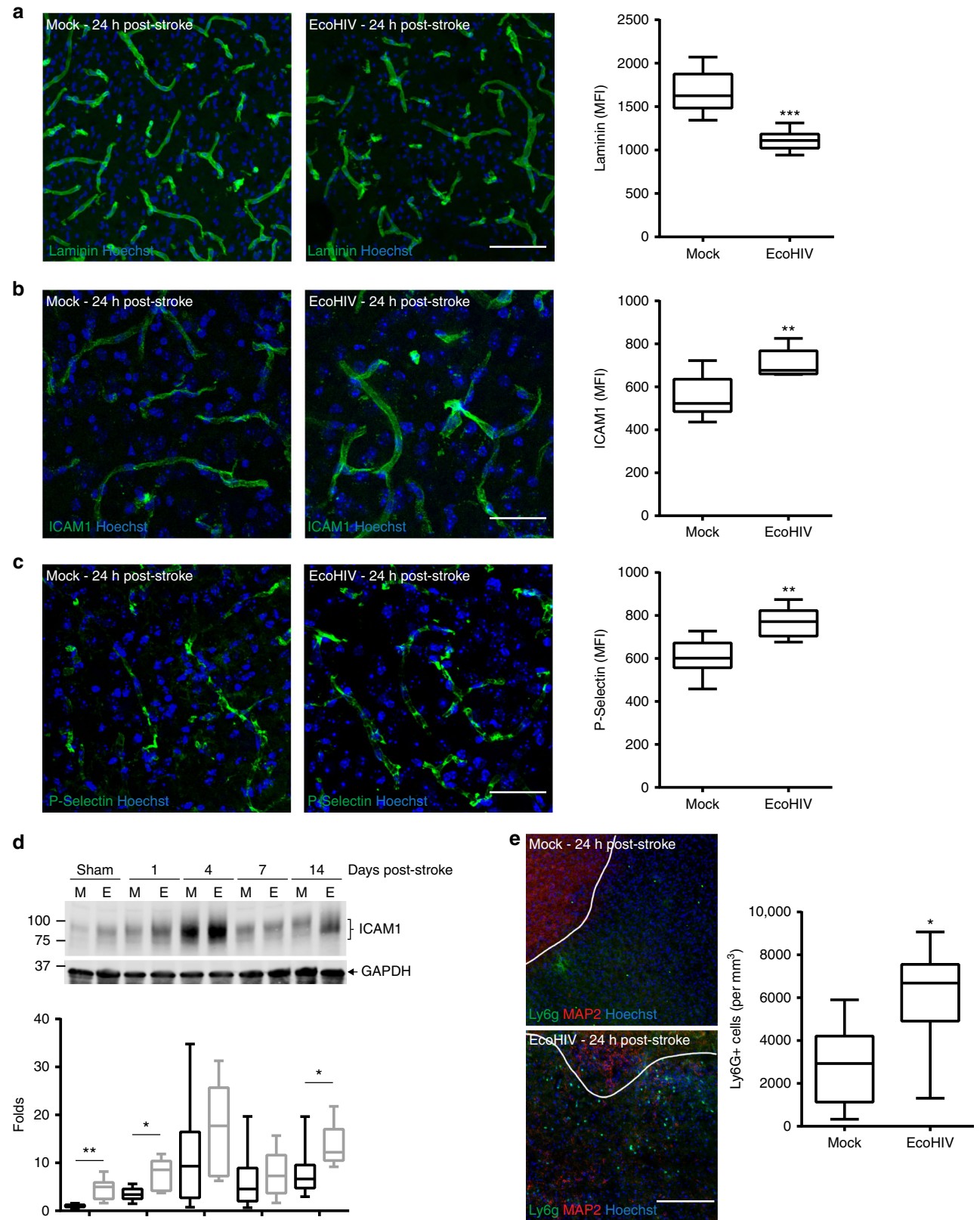

To further analyze the impact of EcoHIV on the kinetics of inflammatory processes following ischemic stroke, we next assessed protein levels of GFAP and Iba1. A significant increase in GFAP in the EcoHIV group was observed at days 4, 7, and 14 post-stroke (Fig. 4g). Moreover, a significant increase in Iba1 was detected in EcoHIV infected mice as compared to mock at 1, 7,

and 14 days post stroke induction (Fig. 4h). These results indicate that the overall inflammatory response following stroke is more pronounced in the brains of EcoHIV infected mice.

**Ischemic stroke results in increased CNS viral load.** Brain infections by HIV in humans in the current ART era are

**Fig. 3** EcoHIV diminishes post-ischemic stroke NVU recovery. Mice were infected with EcoHIV and subjected to stroke as in Fig. 1. Brain sections were stained for laminin (**a**), ICAM-1 (**b**) and P-selectin (**c**) 24 h post stroke, and quantified for mean fluorescence index (MFI); $n = 6$ mice per group, 10 microvessels per mice, 2 independent experiments. **d** Time course of ICAM1 expression levels as quantified by western blotting in sham and ischemic stroke animals at days 1, 4, 7, and 14 post-stroke both in mock (M) and EcoHIV-infected (E) mice. Representative blots are shown, and quantified results from 5–12 samples per group are illustrated on the bar graphs. **e** Representative image (left) and quantified results (right) of infiltration of the infarct area by Lys6g immunoreactive cells (neutrophils) at 24 h post-ischemic stroke. The sections were also stained for MAP2 (neurons) and Hoechst (nuclei). Absence of MAP2 staining indicates infarct area. Data quantified from 6 mice per group, 2 independent experiments, 4 fields of view per mice at ×20 magnification; Z stack images. Whiskers-box plots represent centerline median, with interquartile range and min-max whiskers. Source data are provided as a Source Data file. **p < 0.01; ***p < 0.001; unpaired t test. **a–c** Scale bars: 40 μm; **e** scale bar: 320 μm

characterized by a low level of viremia, a feature that is well represented in the EcoHIV infection model. While such viremia potentiates ischemic stroke outcome (Fig. 1), there is no available data on the impact of stroke on HIV viral load in the brain or periphery. The problem is of high significance because of the relationship between viral activity and proinflammatory responses.

Using our EcoHIV brain infection model, we investigated the impact of ischemic stroke on HIV load both in the CNS and plasma. Most significantly, we detected an increase in viral presence in the brain at 7 days post-stroke (Fig. 5a). In addition, we also observed an increased viral load in the plasma at the same time post-ischemic stroke (Fig. 5b). The increase in the brain appeared to be located primarily at the periphery to the infarct area in clusters of cells positive for HIV protein p24, indicating active viral replication (Fig. 5c). Very few, if any, p24 positive cells were detected in infected but sham treated brains. We next characterized the cells harboring active HIV replication by cell type-specific immunofluorescence. No co-localization of p24 was detected with the astrocyte marker GFAP (Fig. 5d), however positive co-localization was observed between p24 and the markers Iba1 and Tmem119 (Fig. 5e, f). These results indicate that HIV is reactivated in microglia and possibly in monocytes/macrophages as the result of stroke. Thus, tissue injury associated with ischemic stroke can stimulate active HIV replication in infected brains, potentially by reactivation of the virus from CNS reservoirs.

**ART CPE score correlates with post-stroke treatment outcome**. The beneficial impact of ART is undeniable, however, recent studies have highlighted the importance and difficulty of targeting HIV replication in the CNS. On the other hand, there is an ongoing discussion about off target toxicity of ART that can be potentiated by increased delivery into the brain. Therefore, we compared two contrasting HIV therapies in our experimental ischemic stroke model. The first ART combination consisted of Raltegravir, Emtricitabine and Tenofovir and has a CPE score of 7, which is considered low. This drug combination (referred by us as ART-7) is a recommended therapy in HIV treatment naïve patients. In addition, we designed a treatment consisting of Zidovudine, Emtricitabine and Nevirapine that scored 11 on the same scale (named ART-11 by us). Finally, a control group was given saline. We previously published that a drug used in the treatment of HIV, Efavirenz, can increase infarct size[77]. Given this possibility, we first evaluated the impact of ART-7 and ART-11 on ischemic tissue damage in non-infected mice. As indicated in Fig. 6a, both ART formulations did not affect infarct size in infected brains 24 h post-stroke.

We then expanded the model and subjected EcoHIV infected mice to ischemic stroke, followed by two weeks of ART-7 or ART-11 administration, with the control mice receiving a vehicle. EcoHIV infection treated with saline resulted in enhanced infarct size, and both ART-7 and ART-11 exhibited a significant protection as demonstrated by a decrease in tissue damage. At

24 h post-stroke, both ART-7 and ART-11 were equally effective in diminishing the infarct size (Fig. 6b). However, a significant difference between the treatments was observed 7 days post-ischemic stroke, with ART-11 being more potent in decreasing infarct size (Fig. 6c). The time points for these analyses (namely, 24 h and 7 days post-ischemic stroke) were preselected based on the results from Fig. 1. Overall, these results support our hypothesis that more effective targeting of HIV replication in the CNS can be beneficial in ischemic stroke.

**Effective targeting of EcoHIV diminishes brains inflammation**. To dissect how ART-11 can improve stroke outcome, we first analyzed the levels of inflammation markers that were previously found to be upregulated in infected brains. These analyses were performed 7 days post stroke in the ipsilateral hemisphere. A significant reduction in the Iba1 mRNA levels was observed in mice treated with both ART-7 and ART-11 (Fig. 7a). On the other hand, a significant decrease in GFAP mRNA level was detected only in mice that received ART-11 (Fig. 7b).

To confirm these results and to assess a long-term impact of therapy, brain samples were next analyzed for Iba1 (Fig. 7c), GFAP (Fig. 7d), and ICAM1 (Fig. 7e) protein expression by immunoblotting at 14 days post ischemic stroke. The results revealed a significant and sustained elevation in protein levels of these inflammatory markers. Importantly, treatment with ART-7 or ART-11 markedly attenuated these effects. Both ART treatments were equally effective in protection against EcoHIV-associated inflammatory responses.

An anti-inflammatory response is also important in improving tissue recovery. Emerging data indicates that ICAM-5, a member of the ICAM family, can bind LFA-1 and inhibit leukocyte activation[78]. Therefore, it is important to note that a significant increase in ICAM-5 mRNA expression occurred in ART-11 (but not ART-7) treated mice as compared to saline control (Fig. 7f). The precise role of regulatory T cells remains obscure in stroke; however, several groups have shown that they can be beneficial to tissue recovery. Based on these reports, we analyzed the levels of Foxp3, a marker for the regulatory T cells. There was a significant increase in Foxp3 mRNA levels in ART-11 but not in ART-7-treated animals when compared to controls (Fig. 7g). Other important mediators of tissue damage are matrix metalloproteases (MMPs) as they have been linked to exacerbated tissue damage in the CNS, such as the disruption of the BBB, activation of glial cells, and white matter tissue lesions[79]. We detected a significant post-ischemic stroke increase in MMP2 mRNA expression in EcoHIV infected brains as compared to mock-infected mice, an effect that was significantly abrogated in ART-11 (but not in ART-7) treated mice (Fig. 7h). The expression of MMP9 mRNA showed a tendency to be increased in the post-ischemic stroke EcoHIV-infected brain; nevertheless, these changes were not statistically significant (Fig. 7i).

We next evaluated if the impact of ART-7 vs ART-11 treatment on tissue damage and inflammatory responses correlates with alterations of EcoHIV viral presence in the CNS.

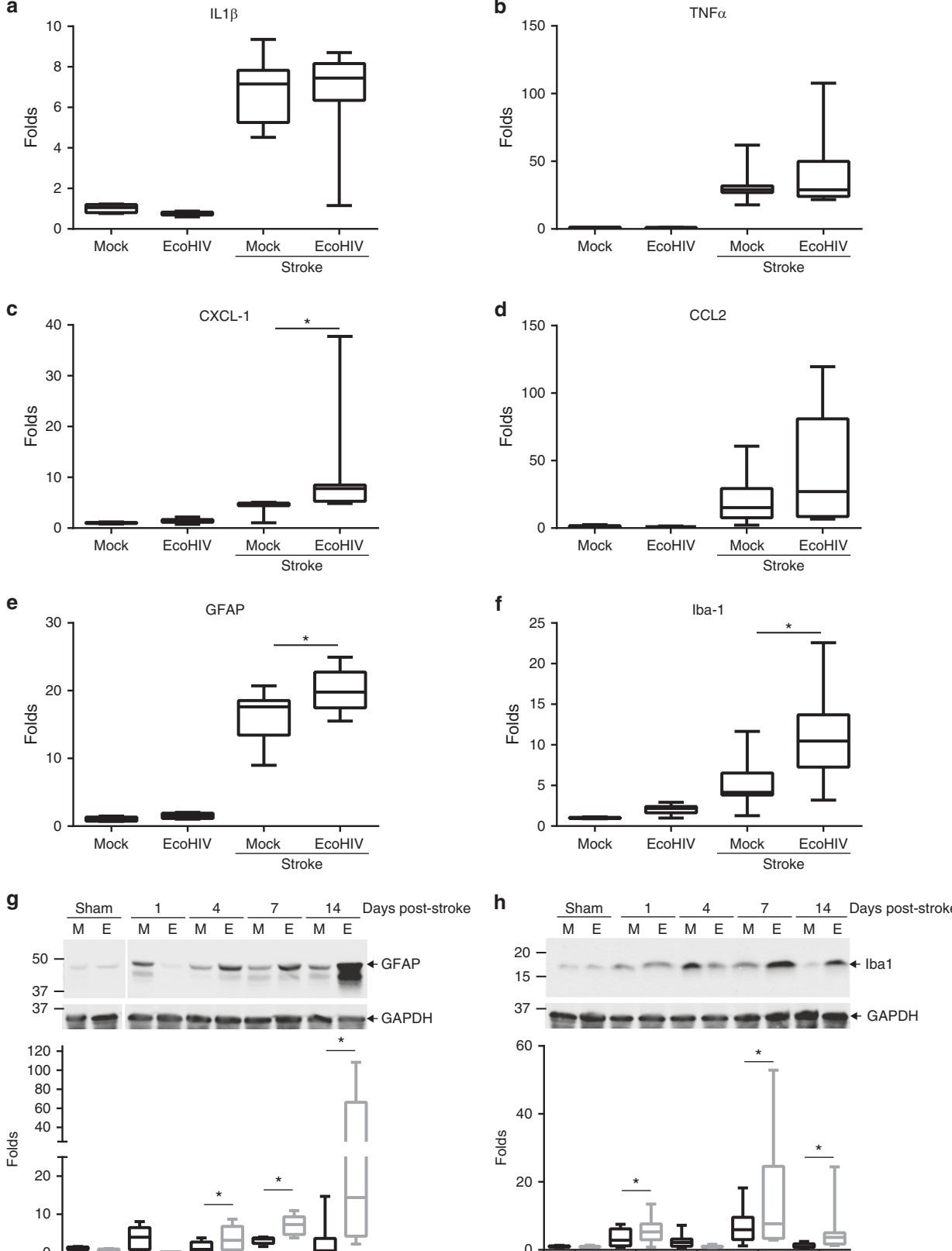

**Fig. 4** Prolonged post-ischemic stroke inflammation in EcoHIV-infected brains. Mice were infected with EcoHIV and subjected to stroke as in Fig. 1. mRNA expression of cytokines IL1β (**a**) and TNFα (**b**) chemokines CXCL1 (**c**) and CCL2 (**d**), and cellular activation markers GFAP (**e**) and Iba1 (**f**) were assessed by real-time qPCR 7 days post stroke. Sham, $n = 6$; Stroke, $n = 12$ mice per group, 3 independent experiments. Time course of GFAP (**g**) and Iba1 (**h**) protein expression levels as quantified by western blotting in sham and ischemic stroke animals at days 1, 4, 7, and 14 post-stroke in mock (M) and EcoHIV-infected (E) mice. Representative blots are shown, and quantified results are illustrated on the bar graphs. $n = 5–12$ mice per group, 3 independent experiments. Whiskers-box plots represent centerline median, with interquartile range and min-max whiskers. Source data are provided as a Source Data file. $*p < 0.05$; one-way ANOVA, followed by Tukey multiple comparison test (**a**–**f**) and unpaired $t$ test (**g**, **h**)

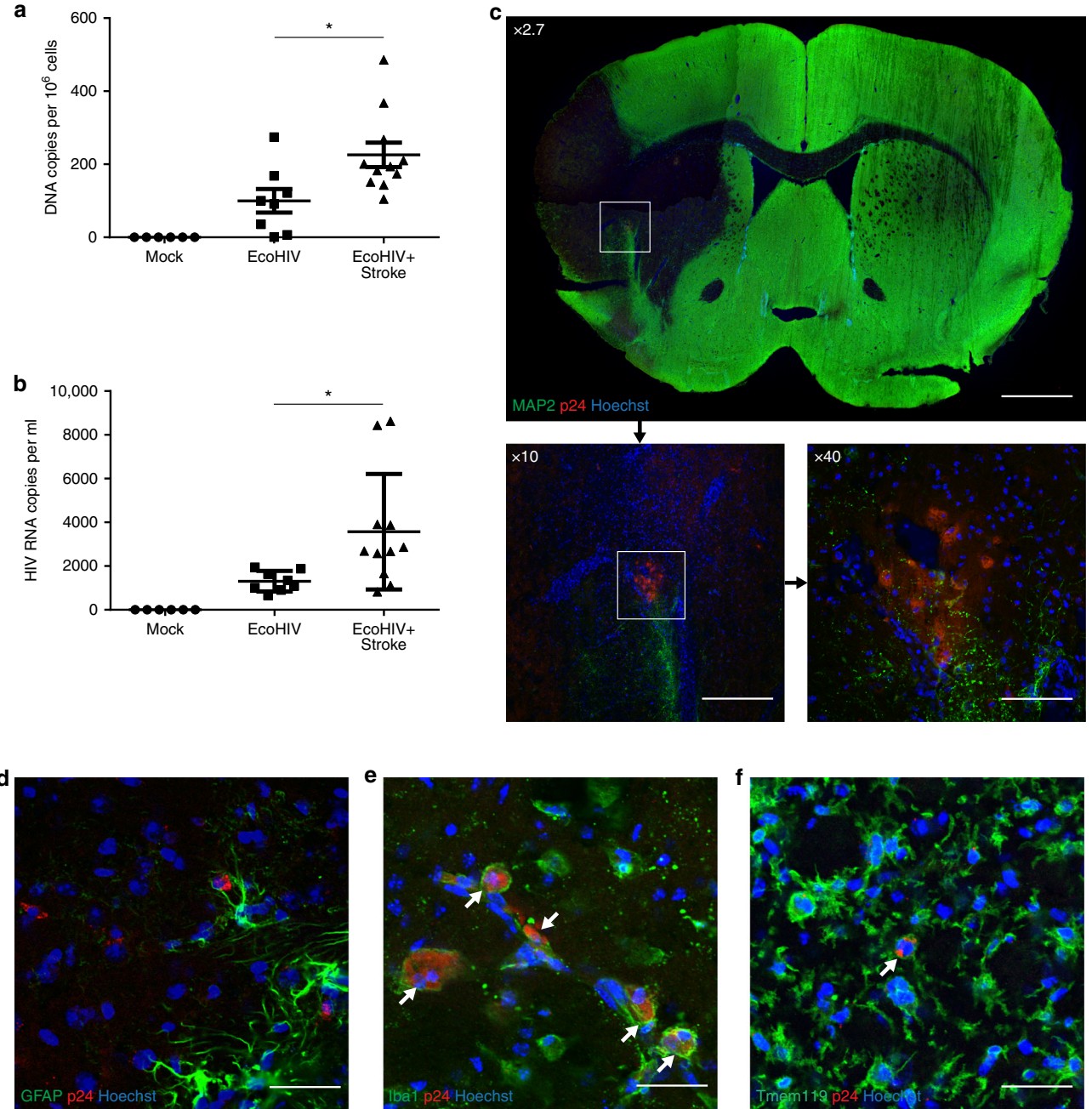

**Fig. 5** Ischemic stroke results in increased CNS viral load. Mice were infected with EcoHIV and subjected to stroke as in Fig. 1. **a** Analysis of EcoHIV DNA genome copies in the ipsilateral hemisphere in mock-infected and EcoHIV-infected mice at day 0 (before stroke) and day 7 post-stroke; $n = 6$–11 mice per group, 2 independent experiments. **b** Quantification of serum viral RNA genome copies in the same mice as in (**a**); $n = 6$–11 mice per group, 2 independent experiments. **c** Brain sections stained for MAP2 (neurons), p24 (HIV marker), and Hoechst (nuclei). Dark area denotes infarct area; scale bar 1 mm. Enlargements (10 ×, scale bar 320 μm; and 40 ×, scale bar 80 μm) of the area with cells positive for HIV p24. In addition, sections were stained with p24 antibody and cell-specific markers to determine the cell type harboring HIV, GFAP (astrocytes, **d**; scale bar 20 μm), Iba1 (monocyte/macrophages/microglia, **e**; scale bar 20 μm), and Tmem119 (microglia, **f**; scale bar 20 μm). Double-stained cells are indicated by arrows. Data presented as mean and SEM with individual data points. Source data are provided as a Source Data file. *$p < 0.05$ vs EcoHIV; unpaired $t$ test

Ischemic stroke markedly reactivated HIV as determined by an increase in HIV DNA copy number in the brain (Fig. 7j), confirming the results reported in Fig. 5. However, both ART-7 and ART-11 decreased these values to non-detectable levels (Fig. 7j). When analyzing the spleens from the same animals, around 50% of the animals treated with either ART-7 or ART-11 had detectable levels of HIV DNA (Fig. 7k). The detection limit for HIV DNA employed in the present study is 50 copies per million cells. Therefore, we cannot preclude that ART-7 and

ART-11-treated mice exhibited differences in HIV CNS viral load since they were both below the sensitivity level of our RT-qPCR assay.

**ART accelerates post-ischemic stroke functional recovery**. To assess whether EcoHIV infection affects post-stroke functional recovery, mice were next analyzed for neurodeficit score and sensorimotor functions. Animals were subjected to Eco-HIV infection, administration of ART, and/or ischemic stroke as in

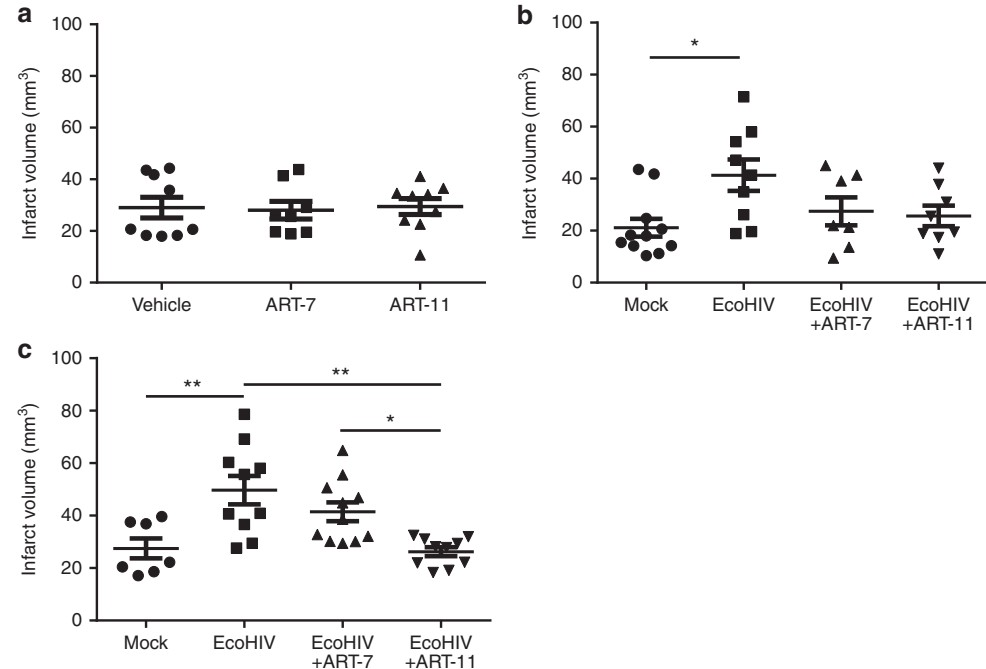

**Fig. 6** ART CPE score correlates with post-stroke treatment outcome. **a** Quantification of infarct size of non-infected mice treated with ART-7 or ART-11. Mice were administered with antiretroviral drugs as ART-7 or ART-11 mixture for 2 weeks. Then, ischemic stroke was induced and infarct volume was assessed 24 h post-stroke. **b**, **c** Mice were infected with EcoHIV as in Fig. 1, and ART-7 or ART-11 was introduced one week later for 2 weeks. Then, ischemic stroke was induced, and infarct volume was evaluated after 24 h (**b**) or 7 days (**c**). ART continued to be administered post-stroke; $n = 7$–11 mice per group, 2 independent experiments. Data presented as mean and SEM. Source data are provided as a Source Data file. *$p < 0.05$ or **$p < 0.01$; one-way ANOVA, followed by Tukey multiple comparison test

Figs. 6 and 7, and behavioral testing was performed at days 1, 4, 7 and 14 post-stroke. Significant differences in neurodeficit score that affects primarily motor functions were detected between mock and Eco-HIV-infected mice at days 1, 7 and 14 post stroke (Fig. 8). EcoHIV-infected mice treated with ART-7 or ART-11 performed similarly to the mock-infected group. The comparison between the EcoHIV and EcoHIV + ART-11 was significantly different at days 1, 7, and 14 post stroke, while significant protection by ART-7 was noted only at the later time points, such as days 7 and 14 post ischemic stroke.

Sensorimotor testing was conducted using the corner test, which has been demonstrated to have a long term sensitivity[80]. This test proved to be very challenging in mice. The mock and the EcoHIV-infected + ART-7 or ART-11 groups clustered together and showed a clear trend of being distinct from EcoHIV-infected mice that were not subjected to an ART regimen. However, a high in-group variability of this assay precluded significantly different results, except for the EcoHIV + stroke group as compared to the sham group (Supplementary Fig. 3). Taken together, behavioral testing indicated that EcoHIV infection can markedly impair post-stroke functional recovery; however, ART can provide protection against post-stroke motor impairment.

## Discussion

Emerging clinical and epidemiological data has indicated a link between HIV infection and stroke; however, lack of suitable animal models hampered mechanistic research in this field. In the current study, we present direct in vivo evidence of the relationship between HIV infection and ischemic stroke. While the model employed in this study relies on the mouse adapted strain of HIV, called EcoHIV, it has been demonstrated by multiple groups that EcoHIV can replicate several steps of normal HIV

infection progression, localizing in similar organs and infecting cells that are normally targeted by HIV[81–83].

Our results demonstrate that EcoHIV infection predispose the brain tissue to ischemic injury as demonstrated by an increase in infarct volume in infected mice 24 h post-stroke induction (Fig. 1b). Importantly, infection also affected tissue recovery, demonstrating that not only early but also late events in ischemic stroke are affected by HIV. Moreover, we linked the association between EcoHIV and ischemic stroke to EcoHIV-mediated disruption of the BBB and vascular pathology that is consistent with small vessel disease. Using several methods we demonstrated that infection with EcoHIV leads to functional and structural alterations of the BBB integrity, highlighted by the reduced levels of major tight junction proteins, such as ZO-1 (Fig. 2a). This impact on the BBB is in line with the observations from our and other groups that demonstrated that the HIV tat protein can affect the expression of several tight junction proteins, and that endothelial permeability is enhanced by HIV infection[84–86]. In the present study, we also demonstrated that HIV infection can lead to an increase in vascular inflammation, emphasized by overexpression of cell adhesion molecules and MMP-2, as well as infiltration of brain tissue with inflammatory cells (Figs. 2c, d, 3b–e, 4, and 7). Taken together, the results reported in the current manuscript indicate that HIV infection results in a cumulative disruption of the BBB, which predisposes to tissue injury and more severe ischemic stroke outcomes. These observations are consistent with the evidence that a compromised BBB may affects the development of CVD, potentially contributing to stroke outcomes[84,87,88].

A high level of brain plasticity in rodents is responsible for the remarkable tissue recovery that is typically observed in mice subjected to stroke. However, our analyses performed 24 h post-stroke demonstrated that the response of the neurovascular unit to restore both BBB integrity and the extracellular matrix was compromised in EcoHIV-infected mice as compared to mock-

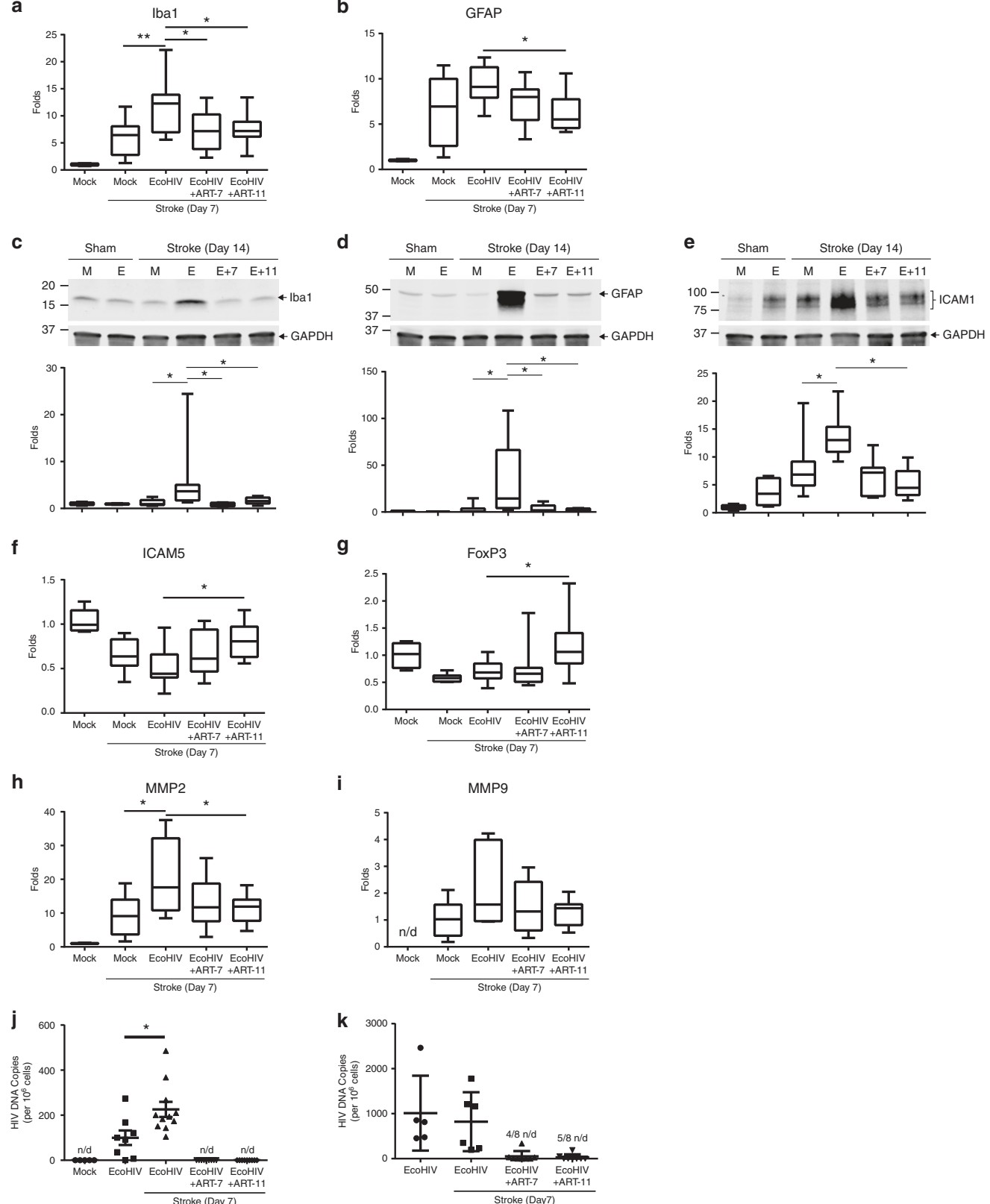

infected controls (Fig. 3). It was previously reported that stroke-induced COX-2[76] can lead to remodeling of the extracellular matrix by upregulating laminin expression, a process that was linked to overexpression of COX-2. Indeed, COX-2 is usually upregulated in endothelial cells exposed to HIV Tat; however, we observed that the association of COX-2 with disruption of the

BBB integrity may only be coincidental as specific inhibition of this enzyme did not protect against HIV Tat-induced BBB breakdown and downregulation of ZO-1[86]. This phenomenon, combined with complex cellular and tissue processes associated with infection may explain the reduced upregulation of laminin in the EcoHIV-infected group as observed in the present study

**Fig. 7** Effective targeting of EcoHIV diminishes post-ischemic stroke inflammation. Mice were infected, treated with ART-7 and ART-11, and subjected to ischemic stroke as in Fig. 6b, c. mRNA levels of cellular activation markers Iba1 (**a**) and GFAP (**b**) 7 days post stroke. Protein expression levels of Iba1 (**c**), GFAP (**d**), and ICAM1 (**e**) were quantified by immunoblotting in sham and ischemic stroke animals at day 14 post-stroke in mock (M) and EcoHIV-infected (E) mice that were treated with ART-7 (E + 7) or ART-11 (E + 11). Representative blots are shown, and quantified results are illustrated on the bar graphs. mRNA levels of anti-inflammatory markers ICAM-5 (**f**) and FoxP3 (**g**), and tissue degrading enzymes MMP2 (**h**) and MMP9 (**i**) were quantified by RT-qPCR. Impact of therapy on viral DNA genome levels was evaluated in ipsilateral hemisphere (**j**) and spleen (**k**); $n = 5$–16 mice per group, 2 independent experiments. Whiskers-box plots represent centerline median, with interquartile range and min-max whiskers. Other graphs represent data as mean and SEM with individual data points. Source data are provided as a Source Data file. $*p < 0.05$ or $**p < 0.01$; one-way ANOVA, followed by Tukey multiple comparison test (**a**, **b** and **f**–**k**), and unpaired $t$ test (**c**–**e**)

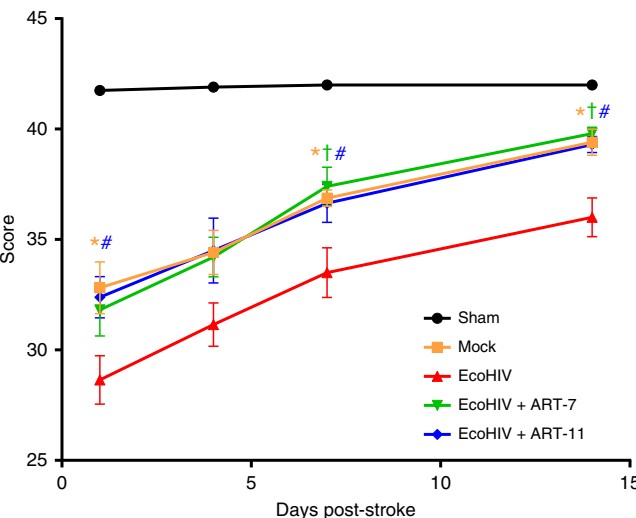

**Fig. 8** ART accelerates post-ischemic stroke functional recovery. Mice were infected, treated with ART-7 and ART-11, and subjected to stroke as in Figs. 6 and 7, followed by evaluation for neurodeficit score at days 1, 4, 7, and 14 post-stroke; $n = 9$–20 mice per group, 2 independent experiments. Data presented as mean and SEM. Source data are provided as a Source Data file. $*p < 0.05$ Mock vs EcoHIV; $†p < 0.05$ EcoHIV + ART-7 vs EcoHIV; $\#p < 0.05$ EcoHIV + ART-11 vs EcoHIV; one-way ANOVA, followed by Tukey multiple comparison test

(Fig. 3a). In addition to disrupted vascular functionality, we observed a markedly elevated expression of cell adhesion molecules. While such an increase was expected in a post-stroke brain and has been consistent with other reports[89,90], it was of great significance that these responses were potentiated in EcoHIV infected mice. Indeed, overexpression of cell adhesion molecules can result in enhanced extravasation of pro-inflammatory cells, increased tissue damage, and can contribute to a no-reflow effect[74]. Consistent with such responses, significantly increased infiltration of neutrophils in infarcted tissue of infected mice as compared to controls were observed in the present study (Fig. 3e).

Several groups have demonstrated that even in patients efficiently treated with ARVds, a low level of HIV activity remains, resulting in the production of several viral proteins that are known to be toxic. The brain may be particularly prone to such responses due to limited penetration of ARVds across the BBB and/or accelerated removal of these drugs from brain tissue. A persistent viral replication can affect several cellular processes, and also induce chronic inflammation, resulting in a sensitization of the tissue to exacerbate injury, i.e., the process that we observed in infected mice upon induction of ischemic stroke. A feature of inflammatory responses is an increase in influx of immune cells into the injury site. Thus, it is of no surprise that a marked increase in several inflammatory markers was observed in post-stroke EcoHIV-infected brains. Stimulation of inflammatory

processes has the potential of increasing viral activity. Indeed, we demonstrated a significant increase in ipsilateral EcoHIV DNA copy number following ischemic stroke. Cells harboring HIV were observed mainly at the periphery of the infarct area, corresponding to activated immune cells. Using several cell markers, we conclusively demonstrated that both microglial cells and monocytes/macrophages expressed HIV-specific p24, which suggest an active infection (Fig. 5). Thus, increased HIV presence in the infected post-stroke brain can originate from both brain-specific reservoirs and from the peripheral infiltration.

In order to design a more informed therapy of HIV infection in the brain, the CPE scale, in which drugs are scored according to their ability to reach the CNS, was recently developed. However, enhanced levels of ARVds in the brain may also induce more pronounced toxicity of these drugs in a delicate brain environment that is prone to neurotoxicity and neuroinflammation. Thus, it has been unclear whether more effective ARVd accumulation in the brain would translate into a more or less beneficial impact on cerebrovascular events in HIV-infected brain. This emerging problem was directly addressed in the current manuscript by using two ART regimen of low and high CPE score; ART-7, and ART-11, respectively. Interesting findings indicate that both treatments, regardless of CPE score, diminished the area of tissue injury within the first 24 h post- ischemic stroke. However, ART-11 was markedly more beneficial in promoting a long-term tissue recovery that was analyzed 7 days post-ischemic stroke. We also demonstrated that this is a long-term effect that is preserved at day 14 post-stroke (Fig. 7c–e). This impact was also well correlated with more efficient protection against inflammatory responses, such as reduced levels of pro- and enhanced levels of anti-inflammatory mediators. In this context, the role of ICAM-5 appears to be of particular interest due to its role in the inhibition of microglial activation[78]. Upregulation of ICAM-5 in ART-11 treated mice may have a dual role in stroke by diminishing both the inflammatory responses and also by being neuroprotective[91]. Similar impact may be exerted by regulatory T cells that were demonstrated to have beneficial role in stroke[92–94]. Neurodeficit score performed as part of our study demonstrated that HIV infection can worsen post-stroke outcomes not only in terms of tissue injury, but also functional motor recovery (Fig. 8). Furthermore, this test supported our findings that ART can improve stroke outcomes in infected mice.

In summary, our results clearly demonstrate that HIV infection has a deleterious impact on ischemic stroke outcome by inducing structural and functional alterations of the BBB that sensitize brain tissue to ischemic injury. Both more pronounced tissue injury immediately after ischemic stroke, and the delay in post-stroke recovery were correlated with enhanced HIV viral load and more robust pro-inflammatory responses. Importantly, the use of ART with a high CPE score resulted in more beneficial impact on post-stroke recovery in HIV-infected brains. Overall, these results provide the first mechanistic link between HIV infection and ischemic stroke outcome, and also demonstrate that efficient targeting of HIV infection in the brain can diminish the

cerebrovascular dysfunction and facilitate post-ischemic stroke tissue repair and functional recovery.

## Methods

**Mice, surgical procedures, and ARVd administration**. We employed 12 weeks old male C57BL/6 J mice (Jackson Laboratories). Animals were housed in an AALAC accredited facility and all animal procedures were conducted in accordance with the protocols approved by the University of Miami Institutional Animal Care and Use Committee (IACUC) and were consistent with National Institutes of Health (NIH) guidelines. Before the experiments, animals were allowed to acclimatize to the housing facility for one week and were given ad labium access to food and water.

EcoHIV infection was performed by infusing the virus (1 µg of p24) via the internal carotid artery[70]. Briefly, a midline neck incision was performed to expose the common carotid artery, along with the branching towards the external and internal carotid arteries. A canula was inserted into the external carotid artery and virus was infused directly into the internal carotid artery. ARVd administration was initiated at day 7 post-infection and was performed by gavage, once a day for 14 days. Treatments consisted of either Vehicle (13.5% DMSO in saline), ART-7 (Raltegravir 6.67 mg/kg, Emtricitabine 3.33 mg/kg, and Tenofovir 5 mg/kg; combined CPE score = 7), or ART-11 (Zidovudine 5 mg/kg, Emtricitabine 3.33 mg/kg, and Nevirapine 3.33 mg/kg; combined CPE score = 11).

Ischemic stroke was induced by the middle cerebral artery occlusion (MCAO) technique 3 weeks post-infection as described in our earlier publications[71]. Briefly, a silicone coated suture (Doccol) was inserted into the common cerebral artery, blocking the blood flow to the middle cerebral artery for 60 minutes. The suture was then removed, allowing for tissue reperfusion. Analyses were conducted at days 1, 4, 7 and/or 14 post stroke. In order to evaluate infarct volume, brains were harvested, sectioned using a 1 mm brain matrix (Braintree Scientific), and stained with 2,3,5-triphenyltetrazolium chloride (TTC, ThermoFisher). The images were captured by a digital camera (Nikon) and infarct volume was calculated using Image J based on the infarct volume and edema correction formula (Infarct size = [ipsilateral volume × infarct volume]/contralateral volume)[95,96].

**HIV production and quantification**. Chimeric HIV virus EcoHIV/NDK was employed for mouse infection. EcoHIV/NDK was generated by replacing the viral gp120 protein with gp80 for the ecotropic moloney murine leukemia virus[82]. The viral plasmid was a kind gift from Dr. David Volsky (Icahn School of Medicine at Mt. Sinai, New York, NY). Viral stocks were prepared by transfecting HEK 293 T/17 cells with the EcoHIV/NDK plasmid using lipofectamine 2000 (Invitrogen). Viral concentration in the filtered supernatant was quantified by p24 ELISA kit (Zeptometrix).

**BBB permeability assay**. BBB permeability was assessed by sodium fluorescein (NaF) extravasation[75] using a dedicated set of mice that were not employed for other analyses. Briefly, mice were injected intraperitoneally with 200 µl of 10% NaF, and euthanized 20 min later. Blood was collected by heart puncture, followed by whole body perfusion with saline. Brains were harvested, hemispheres were homogenized in PBS using tissuelyser (Qiagen), cleared by centrifugation, and protein concentration was quantified by the BCA assay (Pierce). Samples were then precipitated with 100% TCA (Sigma-Aldrich), centrifuged, and supernatants were neutralized with 0.05 M Sodium Tetraborate buffer. Brain and plasma NaF fluorescence was measured at excitation 485 nm and emission 525 nm, and compared to a titration curve for NaF. Extravasation of NaF into the brain was normalized to NaF fluorescence in plasma and sample protein concentration.

**Real-Time PCR**. mRNA from the brain was isolated with RNeasy Lipid Tissue Mini Kit, from the spleen using RNeasy mini Kit and from the plasma using the QIAamp MinElute Virus Spin Kit (All from Qiagen). Reverse transcription and qPCR reaction were performed using the qScript XLT 1-step RT-qPCR ToughMix reagent (Quantabio). Real time PCR was performed using an Applied Biosystem 7500 System, TaqMan Gene Expression Assays, and pre-assembled primers for the indicated genes (CCL2: Mm00441242_m1; CXCL-1: Mm04207460_m1; IL1beta: Mm00434228_m1; TNFalpha: Mm00443258_m1; GFAP: Mm01253033_m1; Iba1: Mm00479862_g1; MMP2: Mm00439498_m1; MMP9: Mm00442991_m1; FoxP3: Mm00475162_m1; ICAM5: Mm00492566_m1) (ThermoFisher). In addition, the GAPDH mRNA level was assessed in duplex for sample normalization. DNA tissue detection of EcoHIV/NDK was performed on harvested tissues that were processed with the QIAamp DNA mini kit (Qiagen).

HIV was assessed using the following primers and probe: NDKgag_F 5′-GAC ATAAGACAGGGACCAAAGG-3′; NDKgag_R 5′-CTGGGTTTGCATTTTGGA CC-3′; NDKgag_Probe 5′-AACTCTAAGAGCCGAGCAAGCTTCAC-3′. Normalization was conducted based on GAPDH for RNA or mouse HBB for DNA: MGBF 5′-CTGCCTCTGCTATCATGGGTAAT-3′, MGBR 5′-TCACTGAGGCTG GCAAAGGT-3′; MGBP_probe 5′-TTAACGATGGCCTGAATC-3′. Standard curves for EcoHIV and mHBB were run in parallel to assay for copy numbers.

**Protein isolation and western blotting**. Euthanized animals were perfused with 12 ml of saline, after which brains were harvested, separated into ipsilateral and contralateral hemispheres, and snap frozen in liquid nitrogen. Samples were lysed in a 500 µl RIPA buffer containing protease and phosphatase inhibitors in combination with a Tissuelyser LT system (Qiagen), followed by centrifugation. Protein concentration was evaluated in the supernatants by the BCA assay (Pierce). Samples were then denatured with Laemmli sample buffer, and western blotting was conducted using TGX 4–20% gradient precast gels (BioRad) and a turboblot transfer system (BioRad). Proteins were targeted with the following antibodies: Iba1 (Abcam ab17884, 1:1000), GFAP (Cell Signaling 12389, 1:1000), ICAM1 (Abcam ab179707, 1:750), and GAPDH (Novus Biologicals NB600–502FR). Detection was performed with corresponding 800cw tagged antibodies (Licor 926–32213; 1:20000) and anti-GAPDH Dylight 680 (Novus Biologicals NB600–502FR), followed by scanning in the Licor CLx system (Licor). Quantification was performed using Image studio software (Licor).

**Immunofluorescence**. Immunostaining was performed on 30 µm brain cryosections[97], which were stained in suspension before attachment on glass slides. Samples were stained with antibodies against laminin (Abcam ab11575, 1:400), ZO-1 (ThermoFisher 61–7300, 1:150), ICAM1 (Abcam ab179707, 1:100), P-selectin (Abcam ab6632, 1:100), Ly6g (Abcam ab25377, 1:75), MAP2 (Cell Signaling 8707 S, 1:400; Abcam ab11267, 1:400), p24 (NIH AIDS Reagent Program 530, 1:250), Iba1 (Abcam ab5076, 1:200), GFAP (Cell Signaling 12389, 1:500), CD31 (Abcam ab24590, 1:150), and Tmem119 (Abcam ab209064, 1:100). Donkey Alexa-488 and -594 secondary antibodies (Invitrogen, 1:500) were used for target detection; Hoechst (ThermoFisher. 1:5000) was employed to stain cell nuclei. Imaging was performed on an Olympus Fluoview 1200 confocal microscope using 40x or oil immersion 60x lenses, and analyzed using ImageJ software. Quantification of signal levels from brain microvessels was conducted as follows: Z-stack images were acquired by confocal microscopy and then projected to a single image using the maximum intensity selection. Microvessel area was selected based on CD31 or Laminin positive straining as a reference marker with image J. Mean signal intensity was then assessed by the measurement of the total pixel fluorescence intensity and divided by the selection area. Ly6g + cell infiltration quantification was performed using the same image acquisition parameters. MAP2 was employed to delineate infarct area. Pictures were taken using a ×20 objective. Sections were co-stained with Ly6g (Abcam). Cells positive for Ly6g were quantified in 4 fields of view acquired by Z-staking for each animal. Z-stack volume was calculated using FV10-ASW viewer (Olympus) and the final results were converted to cells per cubic millimeter.

**Functional recovery/behavioral testing**. Animal neurodeficit was evaluated using a scale adapted from Cuomo et al[98] that assesses animal condition, behavior, and motor functions using 13 evaluation criteria. The Corner test was conducted according to Zhang et al[80]. Briefly, mice were placed midway between 2 cleaned boards placed at a 30 degree angle with a small opening in the bottom corner. This opening encourages mice to move into the corner; however, following vibrissae stimulation, the animal rears up and turns around. This action was recorded and repeated 8 times per animal. Testing was conducted at the same time of the day and in the same room for all groups to reduce variability. The Investigators performing the test were blinded, with groups' identity being revealed only at the end of the study.

**Reporting Summary**. Further information on research design is available in the Nature Research Reporting Summary linked to this article.

## Data availability

The source data underlying Figs. 1 to 8 and Supplementary Figure 3 are provided as a Supplementary Source Data file. All other data supporting the findings of this manuscript are available from the corresponding authors upon reasonable request.

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

## Acknowledgements

Selected antiretroviral drugs were obtained from the NIH AIDS Reagent Program, Pathogenesis and Basic Research Branch, Division of AIDS NIAID. EcoHIV-NDK viral plasmid was obtained from Dr. David J. Volsky, Icahn School of Medicine at Mt. Sinai, New York, NY. This work was supported by the NIH (MH098891, MH072567, HL126559, DA039576, DA044579, and DA040537). We also acknowledge support for the Miami Center for AIDS Research (CFAR) at the University of Miami Miller School of Medicine funded by a grant (P30AI073961) from the NIH, which is supported by the following NIH Co-Funding and Participating Institutes and Centers: NIAID, NCI, NICHD, NHLBI, NIDA, NIMH, NIA, NIDDK, NIGMS, FIC AND OAR. LB was in part supported by an AHA postdoctoral fellowship (16POST31170002).

## Author contributions

L.B. and M.T. participated in research design, wrote or contributed to the writing of the manuscript, and provided funding; L.B., F.M., M.T., A.L. and E.S. conducted experiments; LB performed data analysis and created all figures.

## Additional information

**Competing interests:** The authors declare no competing interests.

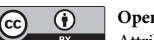 ns license, unless indicated otherwise in a credit line to the material. If material is not included in the article's Creative Commons license and your intended use is not permitted by statutory regulation or exceeds the permitted use, you will need to obtain permission directly from the copyright holder. To view a copy of this license, visit http://creativecommons.org/licenses/by/4.0/.

© The Author(s) 2019

