## [Peer Review File · Nature Communications]

Reviewers' comments:

Reviewer #1 (Remarks to the Author):

The main hypothesis is that low level HIV replication and associated inflammation contribute to the severity of ischaemic stroke. The authors used a mouse model and EcoHIV a virus adapted for use in mice to explore the hypothesis. The authors main claims are:

1. That the size of ischaemic strokes is increased in the presence of HIV infection and post stroke recovery is delayed.
2. That HIV infection results in blood brain barrier disruption prior to ischaemic stroke, with evidence of tight junction disruption rather than disruption of laminin.
3. HIV infection results in increased expression of adhesion molecules which recruit pro-inflammatory cells. These molecules can result in decreased blood flow.
4. EcoHIV infection affects post ischaemic stroke vascular basement membrane recovery and promotes and prolongs inflammation.
5. Ischaemic stroke reactivates HIV in infected brain
6. Antiretroviral agents with a higher CPE (CNS penetration effectiveness) score are more effective at reducing infarct size than agents with a lower CPE score.
7. Ischaemic stroke resulted in increased EcoHIV load in brain tissue.
8. Antiretroviral agents with higher CPE led to reduced viral load in post stroke brain tissue.

This is the first study to use the EcoHIV mouse model to assesses the impact of EcoHIV infection on the blood brain barrier (BBB) and post ischaemic stroke outcome. The study also assessed the impact of antiretroviral drugs with a range of CPE. The findings provide novel information useful to clinicians (stroke physicians, neurologists, infectious disease physicians), pathologists, as well as basic scientists.

Specific areas that require clarification or consideration:

1. This work focuses on ischaemic stroke. The term stroke includes: cerebral haemorrhage, ischaemic stroke and subarachnoid haemorrhage. The term ischaemic stroke would be more

appropriate than 'stroke' in the manuscript, particularly when defining stroke on page 4 (final paragraph).

2. Stroke in HIV infection has been associated with multiple secondary causes. HIV associated vasculopathy i.e. where HIV is the cause of the vascular disease and secondary ischaemic stroke has been recently classified to highlight the varying arterial changes associated with ischaemic stroke in HIV [Benjamin LA et al, *Lancet Neurol.* 2012; 11: 878-90][Benjamin et al The role of HIV-associated Vasculopathy in the etiology of stroke. *J Infect Dis.* 2017 Sep 1; 216 (5): 545-553][Benjamin et al. Arterial ischemic stroke in HIV: defining and classifying etiology for research studies. *Neurol Neuroimmunol Neuroinflamm* 2016 Jun 30; 3(4); e254]. There may well be a variety of pathogenic processes underlying these phenotypes.

In this work the authors refer to the impact of EcoHIV associated blood brain barrier disruption on small vessel disease, and the evidence provided supports this. In clinical studies, small vessel disease is uncommon as a cause of ischaemic stroke [Benjamin et al *Lancet Neurol* above]. Changes in small vessels have been demonstrated in several studies [Connor et al. *Cerebral infarction in adult AIDS patients: observations from the Edinburgh HIV autopsy cohort; Stroke* 2000 Sep; 31(9): 2117-26] with a study from Edinburgh describing pathological changes consistent with blood brain barrier disruption causing 'incomplete' infarction. However, clinical stroke was very uncommon in this cohort.

It would perhaps be useful to acknowledge that there is a range of arterial disease noted in HIV associated vasculopathy, including small vessel, middle sized arteries, and large arteries involved. Evidence provided in this work confirms that EcoHIV causes disruption of the BBB and this likely contributes to poor outcome following ischaemic stroke. The work does not, however, provide evidence that BBB disruption in HIV infection causes the initial ischaemic event.

The last sentence of the first paragraph on page 16: 'These observations are consistent with the evidence that a compromised BBB affects the development and outcome of CVD, including stroke [ref 74, 77, and 78]' is therefore not accurate. It affects the outcome but not clearly the development (it may or may not but the current data does not confirm this).

Furthermore, references 74, 77 and 78 do not support this statement as none focused on the predisposition of BBB disruption to an initial ischaemic event.

3. Some of the concluding sentences are perhaps misleading:

a. The final sentence of the abstract: 'These results provide potential guidelines for treatment of HIV-infected patients that are at risk of developing cerebrovascular episodes.'

The results may provide potential targets for treatment but do not provide guidelines. Furthermore, the data does not provide information relevant to HIV infected patients at risk of developing cerebrovascular episodes, as the data relates to post-ischaemic stroke. The term episodes is also very vague.

b. The final sentence on page 19: 'Overall, and also demonstrate that efficient control of HIV infection in the brain can diminish cerebrovascular comorbidities.'

This sentence is vague and suggests that the data provided shows that efficient control of HIV infection in the brain prevents ischaemic stroke / cerebrovascular disease which it cannot. It may well demonstrate that efficient control of HIV infection post-ischaemic stroke may diminish the extent of the stroke / stroke related complications. The term 'comorbidities' is very vague.

4. It would be useful to have more information about the number of mice used in each component of the study to allow for further studies.

5. Were mice initially used for the BBB studies subsequently included in the post-stroke studies? If yes, could this have influenced the results?

Reviewer #2 (Remarks to the Author):

In their manuscript „Targeting the HIV-infected brain to improve stroke outcome “the authors investigated the effect of HIV infection of stroke outcome. They demonstrated that HIV infection in the brain increases infarct volumes after transient middle cerebral artery occlusion (tMCAO), affects BBB integrity, promotes inflammation and negatively regulates stroke recovery. When animals were treated with combinational retroviral therapies prior to tMCAO, infarct size and inflammatory markers were reduced. Treatment with a combinational antiviral therapy with a higher CNS penetration effectiveness score resulted in a more robust effect.

The underlying question motivating this research is of great interest. To what extent HIV infection is affecting the cardiovascular disease risk is not completely understood and even changing. With new HIV antiretroviral therapies, the organism is facing a chronic persisting low replicating virus that will affect the cardiovascular system to a different extent. To address this question, the authors infected mice with an HIV strain and tMCAO was induced three weeks later. Additional experiments were performed with two different retroviral therapies before tMCAO. The presented results are promising, and the manuscript is well written. However, several major concerns exist:

In general, the presented methods are very limited. The infarct volume (misspelled in every figure) was measured after TTC staining and immunohistochemical and real-time analysis were performed.

-Functional testing of stroke outcome is missing. Behavioral tests or at least a neurological score would underline the clinical relevance of the observed changes.

-Kinetics are mostly not performed. In figure 1 four different time points are analyzed (d1,4,7,14). These are common intervals to address stroke development and recovery. Unfortunately, the following analyses only addresses one timepoint for each experiment. Given the limited data, the development cannot be extrapolated.

-The manuscript contains many representative images and experiments (e.g. fig. 5C-F) that are not quantified, or they do not represent every timepoint (e.g. fig 1A).

-While the key findings are interesting in principle there is a significant lack of mechanistic data and the paper in its present form stays rather descriptive.

-In the present paper stroke outcome was defined as damaged area determined via TTC staining, mRNA levels of a limited number of genes and representative immunohistochemical pictures (not all of them were quantified). To underline their statement, kinetics need to be performed (d1 to d14) to understand the changes responsible for the described increase of infarct volume in EcoHIV infected

mice. Additionally, stroke outcome is not only defined by infarct area. To define the improved stroke outcome functional tests like Bederson score or behavioral tests are mandatory, but missing. Inflammatory processes are mentioned and addressed by some quantitative PCR data, but cellular infiltration was neither quantified nor composition (and especially changes in composition) addressed in detail with flow cytometry or systematic immunohistochemical quantification (Granulocytes, Iba+ and Tregs are already mentioned in the manuscript at some point). After ART treatment the decreased tissue damage is solely explained via changes in mRNA levels of some genes, but cellular infiltration and behavioral outcome/functional tests are not addressed at all. Taken together the mechanism underlying the observed effects on infarct volume is not analyzed in detail.

Minor points:

-Fig. 1: How was TTC staining quantified?

-Fig. 2: How are vessels identified? In the text costaining with CD31 is mentioned but not shown. Additionally, the sequence of figure 2C to E is mixed up in the results and in the figure legend. Unfortunately, only one time point is analyzed.

-Fig. 3: In the text it is stated: "while stroke alters the architecture and the inflammation of the CNS..." This is an overstatement. The manuscript would clearly benefit if cellular infiltrations after stroke are determined and quantified. In general, I would suggest combining figure 2 and 3 and represent the data similar to figure 7. Then accessibility of data would be improved. Only one time point is analyzed.

-Fig. 4: The control group of EcoHIV without stroke is missing. qPCR analyses would be underlined if at least some of the genes were also quantified on protein level. Only one time point is analyzed. Please specify primers or assays used for qPCR.

-Fig. 5: Significance is different to identify. It is easier accessible if presented with bars comparable to figure 1 or 6. If EcoHIV +/- stroke group in figure 5A and 7G is the same, different significances are depicted (5A: not significant, 7G: significant). Only one time point is analyzed.

-Fig. 6: Only two time point are analyzed.

-Fig. 7: Only one time point is analyzed.

Answers to comments made by Reviewers

We thank Reviewer 1 for the helpful comments and suggestions.

Answers to comments made by Reviewer 1.

The main hypothesis is that low level HIV replication and associated inflammation contribute to the severity of ischaemic stroke. The authors used a mouse model and EcoHIV a virus adapted for use in mice to explore the hypothesis. The authors main claims are:

1. That the size of ischaemic strokes is increased in the presence of HIV infection and post stroke recovery is delayed.
2. That HIV infection results in blood brain barrier disruption prior to ischaemic stroke, with evidence of tight junction disruption rather than disruption of laminin.
3. HIV infection results in increased expression of adhesion molecules which recruit pro-inflammatory cells. These molecules can result in decreased blood flow.
4. EcoHIV infection affects post ischaemic stroke vascular basement membrane recovery and promotes and prolongs inflammation.
5. Ischaemic stroke reactivates HIV in infected brain
6. Antiretroviral agents with a higher CPE (CNS penetration effectiveness) score are more effective at reducing infarct size than agents with a lower CPE score.
7. Ischaemic stroke resulted in increased EcoHIV load in brain tissue.
8. Antiretroviral agents with higher CPE led to reduced viral load in post stroke brain tissue.

This is the first study to use the EcoHIV mouse model to assesses the impact of EcoHIV infection on the blood brain barrier (BBB) and post ischaemic stroke outcome. The study also assessed the impact of antiretroviral drugs with a range of CPE. The findings provide novel information useful to clinicians (stroke physicians, neurologists, infectious disease physicians), pathologists, as well as basic scientists.

Answer. We thank Reviewer 1 for recognizing the strengths of our study and endorsing our manuscript.

Specific areas that require clarification or consideration:

Item 1. This work focuses on ischaemic stroke. The term stroke includes: cerebral haemorrhage, ischaemic stroke and subarachnoid haemorrhage. The term ischaemic stroke would be more appropriate than 'stroke' in the manuscript, particularly when defining stroke on page 4 (final paragraph).

Answer. We fully agree with the Reviewer; indeed, the title of the manuscript was modified to indicate that the study is on ischemic stroke and not on other types of stroke. As suggested by Reviewer 1, the text was corrected throughout the manuscript to clarify the relationship to ischemic stroke. Examples include Abstract, Figure Legends, and multiple changes in the text; e.g., page 7, paragraph 2, page 9, paragraphs 2 and 3, etc.

Item 2. Stroke in HIV infection has been associated with multiple secondary causes. HIV associated vasculopathy i.e. where HIV is the cause of the vascular disease and secondary ischaemic stroke has been recently classified to highlight the varying arterial

changes associated with ischaemic stroke in HIV [Benjamin LA et al, Lancet Neurol. 2012; 11: 878-90][Benjamin et al The role of HIV-associated Vasculopathy in the etiology of stroke. J Infect Dis. 2017 Sep 1; 216 (5): 545-553][Benjamin et al. Arterial ischemic stroke in HIV: defining and classifying etiology for research studies. Neurol Neuroimmunol Neuroinflamm 2016 Jun 30; 3(4); e254]. There may well be a variety of pathogenic processes underlying these phenotypes.

In this work the authors refer to the impact of EcoHIV associated blood brain barrier disruption on small vessel disease, and the evidence provided supports this. In clinical studies, small vessel disease is uncommon as a cause of ischaemic stroke [Benjamin et al Lancet Neurol above]. Changes in small vessels have been demonstrated in several studies [Connor et al. Cerebral infarction in adult AIDS patients: observations from the Edinburgh HIV autopsy cohort; Stroke 2000 Sep; 31(9): 2117-26] with a study from Edinburgh describing pathological changes consistent with blood brain barrier disruption causing 'incomplete' infarction. However, clinical stroke was very uncommon in this cohort.

It would perhaps be useful to acknowledge that there is a range of arterial disease noted in HIV associated vasculopathy, including small vessel, middle sized arteries, and large arteries involved. Evidence provided in this work confirms that EcoHIV causes disruption of the BBB and this likely contributes to poor outcome following ischaemic stroke. The work does not, however, provide evidence that BBB disruption in HIV infection causes the initial ischaemic event.

The last sentence of the first paragraph on page 16: 'These observations are consistent with the evidence that a compromised BBB affects the development and outcome of CVD, including stroke [ref 74, 77, and 78]' is therefore not accurate. It affects the outcome but not clearly the development (it may or may not but the current data does not confirm this).

Furthermore, references 74, 77 and 78 do not support this statement as none focused on the predisposition of BBB disruption to an initial ischaemic event.

Answer. Our study does not evaluate the causative impact of HIV infection on the induction of ischemic stroke but rather the impact of infection on post-stroke outcomes and recovery. This was emphasized in the Abstract and clarified in the revised manuscript (page 6, paragraph 2). The vast majority of presented data (Figures 1, 3-8) refers to post-stroke outcomes, vascular/BBB damage, and post-stroke recovery.

Epidemiological studies link HIV infection to CVD. In addition, recent evidence supports a more prominent connection between HIV and stroke. Several of these studies are referenced in the Introduction. In addition to viral-specific factors, HIV-infected patients exhibit several well-known risk factors for stroke, such as dyslipidemia, large-vessel vasculopathy, and atherosclerosis. This information was added and/or clarified in the revised manuscript (page 4, paragraph 2). The references were also updated and corrected. More details on HIV-related vasculopathy were added to the Introduction section of the revised manuscript (page 4, paragraph 2).

The original sentence on page 16 was reworded to clarify that there is a link between a disrupted BBB and the development of CVD that can affect stroke outcome; specifically, a reference to stroke susceptibility was removed (page 18, paragraph 1).

Item 3. Some of the concluding sentences are perhaps misleading:

a. The final sentence of the abstract: 'These results provide potential guidelines for treatment of HIV-infected patients that are at risk of developing cerebrovascular episodes.'

The results may provide potential targets for treatment but do not provide guidelines. Furthermore, the data does not provide information relevant to HIV infected patients at risk of developing cerebrovascular episodes, as the data relates to post-ischaemic stroke. The term episodes is also very vague.

Answer. We fully agree with the Reviewer. The last sentence of the abstract was changed to be more in line with the findings of this study. This sentence now reads "These results provide potential insight for treatment of HIV-infected patients that are at risk of developing cerebrovascular disease, such as ischemic stroke." The concluding statements (page 21, paragraph 2) were also modified.

We also agree with Reviewer 1 that our study does not provide treatment guidelines, and such a statement was removed from the revised Abstract. Our results demonstrate an advantage of employing a therapeutic strategy based on the use of antiretroviral therapeutics with high CNS penetration efficiency in post-stroke management. Nevertheless, future clinical studies are needed to implement this suggested strategy in patients.

b. The final sentence on page 19: 'Overall, and also demonstrate that efficient control of HIV infection in the brain can diminish cerebrovascular comorbidities.'

This sentence is vague and suggests that the data provided shows that efficient control of HIV infection in the brain prevents ischaemic stroke / cerebrovascular disease which it cannot. It may well demonstrate that efficient control of HIV infection post-ischaemic stroke may diminish the extent of the stroke / stroke related complications. The term 'comorbidities' is very vague.

Answer. We agree with the Reviewer. The sentence was rephrased in the revised manuscript to avoid concerns raised by the Reviewer. The sentence now reads: "Overall, these results provide the first mechanistic link between HIV infection and ischemic stroke outcome, and also demonstrate that efficient targeting of HIV infection in the brain can diminish the cerebrovascular dysfunction and facilitate post-ischemic stroke tissue repair and functional recovery" (page 21, paragraph 2).

Item 4. It would be useful to have more information about the number of mice used in each component of the study to allow for further studies.

Answer. The number of mice per group per experiment was added to the revised Legend for each Figure.

Item 5. Were mice initially used for the BBB studies subsequently included in the post-stroke studies? If yes, could this have influenced the results?

Answer. Mice used for the BBB permeability study were not included in other post-stroke evaluation studies. This was indicated in the revised manuscript (page 23, paragraph 3).

We thank Reviewer 2 for the helpful comments and suggestions.
Answers to comments made by Reviewer 2.

Item 1. Functional testing of stroke outcome is missing. Behavioral tests or at least a neurological score would underline the clinical relevance of the observed changes.

Answer. In response to the Reviewers' comments, we performed an extensive series of experiments evaluating post-stroke functional recovery. The results from these experiments were described on pages 15-16, and included in new Figure 8 and new Supplementary Figure 3. The methods were added to the Methods section of the revised manuscript (pages 26-27, section "Functional recovery/behavioral testing").

Item 2. Kinetics are mostly not performed. In figure 1 four different time points are analyzed (d1,4,7,14). These are common intervals to address stroke development and recovery. Unfortunately, the following analyses only addresses one timepoint for each experiment. Given the limited data, the development cannot be extrapolated.

Answer. In response to the Reviewer's comment, we performed extensive studies to assess the kinetics of several important inflammatory markers, notably Iba1, GFAP, and ICAM1. The results of these series of experiments are described in the revised Results section (page 9, paragraphs 2 and 3; page 10 paragraph 3; page 13 paragraphs 1 and 3), and presented in new Figures 3D, 4G, 4H, 7C, 7D, and 7E.

Item 3. The manuscript contains many representative images and experiments (e.g. fig. 5C-F) that are not quantified (HIV co-staining with HIV infected cells), or they do not represent every timepoint (e.g. fig 1A).

Answer. Quantitative data on HIV infection is presented in Figures 5A, 5B, and 6A-6C. On the other hand, HIV-positive co-staining was not quantified on Figures 5C-5F as the images presented in the manuscript represent only the cell types that are positive for HIV and their localization relative to the infarct area.

Figure 1A does not illustrate each time point due to the complexity of such presentation. Given that the quantitative results from each time point are illustrated in Figure 1B, we feel that this presentation format is friendlier to the readers. In response to the Reviewer's comment, the images of ischemic stroke lesions from all time points and experimental groups were added as Supplementary Figure 1 to the revised manuscript. A reference to this Supplementary Figure was added to the revised manuscript (page 7, paragraph 1).

Item 4. While the key findings are interesting in principle there is a significant lack of mechanistic data and the paper in its present form stays rather descriptive.

Answer. We believe that this manuscript provides the first mechanistic description of the interaction between HIV infection, small vessel disease, and ischemic stroke. We detail the impact of HIV infection on the BBB and small vessel disease before and after the induction of ischemic stroke. HIV infection leads to an increase in the ensuing immune responses, and an increase in viral presence in the CNS and in the circulation, but not in the spleen. The latter finding suggests that ischemic stroke leads to HIV reactivation in latent CNS reservoirs, followed by spreading into the periphery. Finally, our employment of antiretroviral therapy confirms the contribution of HIV to this

process, and that antiretroviral therapy with more sufficient penetration to the CNS can better alleviate post-stroke CNS inflammation. These are all novel and innovative findings that were never reported in the literature, and the employment of antiretroviral therapy makes our studies mechanistic.

Item 5. In the present paper stroke outcome was defined as damaged area determined via TTC staining, mRNA levels of a limited number of genes and representative immunohistochemical pictures (not all of them were quantified). To underline their statement, kinetics need to be performed (d1 to d14) to understand the changes responsible for the described increase of infarct volume in EcoHIV infected mice.

Answer. As indicated in our answer to Item 2 (major points), we evaluated the kinetics of several important inflammatory markers (notably Iba1, GFAP, and ICAM1) in the revised manuscript. The results of these extensive series of experiments are described in the Results section (page 9, paragraphs 2 and 3; page 10 paragraph 3; page 13 paragraphs 1 and 3) of the revised manuscript, and presented in new Figures 3D, 4G, 4H, 7C, 7D, and 7E.

Item 6. Additionally, stroke outcome is not only defined by infarct area. To define the improved stroke outcome functional tests like Bederson score or behavioral tests are mandatory, but missing.

Answer. We agree with the Reviewer. As indicated in our answer to Item 1 (major points), we performed an extensive series of experiments evaluating post-stroke functional recovery. Behavioral evaluation based neurodeficit score and sensorimotor evaluation was added to the revised manuscript. The results from these experiments were described on pages 15-16 of the revised manuscript, and included in new Figure 8 and Supplementary Figure 3. The methods were added to the revised Methods section (pages 26-27, section “Functional recovery/behavioral testing”).

Item 7. Inflammatory processes are mentioned and addressed by some quantitative PCR data, but cellular infiltration was neither quantified nor composition (and especially changes in composition) addressed in detail with flow cytometry or systematic immunohistochemical quantification (Granulocytes, Iba+ and Tregs are already mentioned in the manuscript at some point).

Answer. In response to the Reviewer’s comments, Ly6G+ staining was quantified in the revised manuscript at 24 h post stroke to evaluate infiltration by neutrophils. These results are described on page 9 (paragraph 3) and presented in revised Figure 3E. We also added post-stroke quantification of markers for activated astrocytes (GFAP) and monocytes/macrophages/microglia (Iba1) at several time points as new Figures 4G and 4H. These approaches are described on page 9 (paragraph 3) of the revised manuscript.

Item 8. After ART treatment the decreased tissue damage is solely explained via changes in mRNA levels of some genes, but cellular infiltration and behavioral outcome/functional tests are not addressed at all. Taken together the mechanism underlying the observed effects on infarct volume is not analyzed in detail.

Answer. In addition to inflammatory gene activation, data on HIV levels in treated animals was also presented. In response to the Reviewer's comments, we added expression levels of important inflammatory markers (GFAP, Iba1, and ICAM1) as new Figures 7C, 7D, and 7E. These approaches are described on page 13 (paragraph 3) of the revised manuscript. Finally, we added behavioral data on neurodeficit scoring and sensorimotor evaluation for all groups at all time points previously analyzed (new Figure 8 and new Supplementary Figure 3).

Minor points:

Item 1. Fig. 1: How was TTC staining quantified?

Answer. Additional details of the procedure were added to the Methods section to better describe the quantification process (page 23, paragraph 1).

Item 2. Fig. 2: How are vessels identified? In the text costaining with CD31 is mentioned but not shown. Additionally, the sequence of figure 2C to E is mixed up in the results and in the figure legend. Unfortunately, only one time point is analyzed.

Answer. CD31 staining was not shown to simplify the figures in order to emphasize HIV-induced alterations. In response to the Reviewer's comment, CD31 images in individual panels were added as Supplementary Figure 2 to the revised manuscript. A reference to this Figure was added to the revised manuscript (page 8, paragraph 1).

Label mismatches were corrected; we apologize for this mistake. As indicated earlier, we performed a kinetics study on ICAM 1 expression levels for all the time points tested. The results from these analyses are presented in new Figure 3D.

Item 3. Fig. 3: In the text it is stated: "while stroke alters the architecture and the inflammation of the CNS...." This is an overstatement. The manuscript would clearly benefit if cellular infiltrations after stroke are determined and quantified. In general, I would suggest combining figure 2 and 3 and represent the data similar to figure 7. Then accessibility of data would be improved. Only one time point is analyzed.

Answer. We agree with the Reviewer. As indicated in our response to Item 7 (major points), Ly6G+ staining was quantified in the revised manuscript at 24 h post stroke to evaluate infiltration by neutrophils (revised Figure 3E).

In our original manuscript, the sentence pointed out by the Reviewer is in reference to the NVU (neurovascular unit) and not the CNS. In the revised manuscript, the sentence was modified and the "NVU" was changed to the "BBB." The sentence now reads "Overall, the results in Fig. 3 demonstrate that while ischemic stroke alters the architecture and inflammation of the BBB, infection with EcoHIV can further compromise cerebral vascular responses and exacerbate tissue damage" (page 10, paragraph 1).

Figures 2 and 3 report the results before and after stroke, respectively. Therefore, we prefer to keep these Figures separately.

Item 4. Fig. 4: The control group of EcoHIV without stroke is missing. qPCR analyses would be underlined if at least some of the genes were also quantified on protein level. Only one time point is analyzed. Please specify primers or assays used for qPCR.

Answer. The EcoHIV group was added to the graphs in the revised Figures 4A-4F. We also added Western blot results to illustrate the protein expression of Iba1 and GFAP (i.e., the proteins corresponding to the most significantly altered genes) at all the tested time points (new Figures 4G and 4H). Primers used were pre-assembled assays from Applied Bioscience. This information was added to the revised manuscript and the catalog numbers were added to the revised Methods section (page 24, section "Real-Time PCR").

Item 5. Fig. 5: Significance is different to identify. It is easier accessible if presented with bars comparable to figure 1 or 6. If EcoHIV +/- stroke group in figure 5A and 7G is the same, different significances are depicted (5A: not significant, 7G: significant). Only one time point is analyzed.

Answer. The EcoHIV +/- stroke group in Figures 5A and 7G are the same, and they were marked as significant compared to the controls on both original Figures. However, the asterisk indicating significance could easily be confused with a data point. To remediate this issue, we changed how asterisks were represented on the revised graphs. The additional time point (namely, 14 days post-stroke) was added to new Figures 7C-E.

Item 6. Fig. 6: Only two time point are analyzed.

Answer. We concentrated on these two time points because they were the only time points at which significant differences were observed between the mock and EcoHIV groups in Figure 1. This issue was clarified in the revised manuscript (page 13, paragraph 1).

Item 7. Fig. 7: Only one time point is analyzed.

Answer. Given that the previous Figure showed significant difference between the treatment groups at day 7, we focused on that time point. However, western blot results for the additional time point (namely, 14 days post stroke) were added to the revised Figures 7C-E.

REVIEWERS' COMMENTS:

Reviewer #1 (Remarks to the Author):

Thank you very much for addressing all the points I raised.

Myles Connor

Reviewer #2 (Remarks to the Author):

The revised manuscript has improved substantially. The concerns raised by the reviewer are sufficiently addressed and the added results underline the drawn conclusions. I recommend it is now suitable for publication.